# Commercially Available Enteric Empty Hard Capsules, Production Technology and Application

**DOI:** 10.3390/ph15111398

**Published:** 2022-11-13

**Authors:** Aleš Franc, David Vetchý, Nicole Fülöpová

**Affiliations:** Department of Pharmaceutical Technology, Faculty of Pharmacy, Masaryk University, 612 42 Brno, Czech Republic

**Keywords:** hard capsules, commercial capsules, enteric administration, acid resistance, encapsulation, ECDDT technology

## Abstract

Currently, there is a growing need to prepare small batches of enteric capsules for individual therapy or clinical evaluation since many acidic-sensitive substances should be protected from the stomach’s acidic environment, including probiotics or fecal material, in the fecal microbiota transplantation (FMT) process. A suitable method seems to be the encapsulation of drugs or lyophilized alternatively frozen biological suspensions in commercial hard enteric capsules prepared by so-called Enteric Capsule Drug Delivery Technology (ECDDT). Manufacturers supply these types of capsules, made from pH-soluble polymers, in products such as AR Caps^®^, EnTRinsic^TM^, and Vcaps^®^ Enteric, or capsules made of gelling polymers that release their content as the gel erodes over time when passing through the digestive tract. These include DRcaps^®^, EMBO CAPS^®^ AP, BioVXR^®^, or ACGcaps™ HD. Although not all capsules in all formulations meet pharmaceutical requirements for delayed-release dosage forms in disintegration and dissolution tests, they usually find practical application. This literature review presents their composition and properties. Since ECDDT is a new technology, this article is based on a limited number of references.

## 1. Introduction

In 1982, only 17.5% of newly licensed products were presented as hard gelatin capsules. By 1996, the percentage had already reached 34% [1]. In 2001, capsules accounted for approximately 20% of all prescriptions dispensed [2]. Thus, technically, is it possible to produce an oral solid dosage form containing a drug with limited stability, poor compressibility, or inappropriate organoleptic properties. Most patients also prefer capsules due to their easy swallowing and inert taste [3]. The drug is encapsulated here as a mixture of different excipients to the hard capsule, consisting of a lower part (body) into which the mixture is filled and closed with an upper part of the capsule (cap). Laboratory machines (manual capsule filling machines) [4,5] or semi-automatic alternatively automatic devices (capsule filling machines) [1,3,6] are usually used for encapsulation in pharmacies or research fields. Materials used for manufacturing both capsule parts are often dried glycerogelatin or, more recently, cellulose derivates, mostly hydroxypropyl methylcellulose (HPMC) or other derivatives, and different hydrophilic polymers. At the same time, natural polymers created without using genetically modified organisms (GMOs) and preservatives are being sought [3,7]. There is also a growing demand for new materials in terms of functionality, enabling the dosage form to be delayed, prolonged, or pulsed release [8,9]. In addition, with the development of the administration of locally acting drugs (e.g., for the treatment of Crohn’s disease or colorectal cancer) or probiotics, polymers have been used to protect the dosage forms and their contents as they pass through the gastrointestinal tract (GIT) to its distal parts [10,11]. Recently, there has been an effort to produce enteric capsules in laboratory conditions for small groups of patients, especially in clinical or preclinical trials. These capsules could also be used in fecal microbiota transplantation (FMT), where the damaged intestinal microbiome of the patient is replaced with modified, frozen, or lyophilized stool from a suitable donor (or with lyophilized bacterial cultures from stool). FMT would otherwise have to be performed with nasojejunal tubes or enemas [12,13,14]. As transplants are prepared directly in hospital wards, there is a need to ensure that the process is reproducible, individualized, and feasible without special equipment [13,15]. Coating hard capsules with encapsulated content using enteric polymers is the most common way to achieve acid resistance of pharmaceutical drug form. In clinical practice, it is impossible to use industrial equipment (coaters), but simply immersing the capsules in the polymer dispersion (immersion method) was offered, which often yielded not satisfactory results [16]. In the last 10 years, several manufacturers have focused on producing empty enteric capsules without requiring subsequent coating for simplification and reproducibility of the process [17]. Therefore, Enteric Capsule Drug Delivery Technology (ECDDT) was created, where pharmaceutically approved enteric polymers are incorporated into the capsule shell to provide acid resistance without another coating. This review article describes several commercially available hard capsules prepared by the ECDDT method and their properties [18].

## 2. Enterosolvency of Solid Dosage Forms

Legislation provides synonyms such as delayed-release, gastro-resistant, or enteric-coated for oral enteric dosage forms that resist the acidic environment in the stomach, and then dissolve in the unified simulated environment of the small intestine [19].

Average fasting stomach pH values range from 1.5 to 1.9, but due to inter- and intra- individual variability in the human population, they could be less than 1.0 or even 5.0–6.0. Therefore, after a meal, values could shift from 3.0 to 7.0. The small intestine consists of the duodenum, the jejunum, and the ileum. The duodenum pH values range from 6.0 to 6.5, the jejunum around 6.8, and the ileum up to 7.4 [20,21]. Pharmacopoeial and industry guidelines for enteric capsules require two specific tests to verify acid resistance of drug form, namely disintegration and dissolution. The common principle is to expose the capsule to an acidic and then neutral environment.

For disintegration, the capsules must not show signs of disintegration or crack in simulated gastric juice (SGF) with pH 1.2, within 60 to 120 min (according to requirements of different pharmacopeias), usually without the use of ‘disks’. Subsequently, capsules must disintegrate in simulated intestinal juice (SIF), with pH 6.8, usually with ‘disks’, within 60 min to meet the pharmacopeia limits [22]. Disks are used when specified to prevent floating capsules in the disintegration medium [19]. For the dissolution test, the capsules could release a maximum of 10% content in SGF within 120 min under defined conditions and subsequently a minimum of 80% in pH 4.5–7.5 buffer, usually within 120 min [19,23]. There are also several slightly different conditions and limits across pharmacopeias, presented in Table 1 [22,24,25,26].

## 3. Enteric Polymers

For enteric coating or the ECDDT formation of pharmaceutical dosage form, pharmaceutical technology offers so-called pH-dependent and gel-forming polymers, or a combination thereof.

### 3.1. pH-Dependent Enteric Polymers

Due to their acidic nature, these polymers prevent the drug’s dissolution in the stomach’s acidic environment, but allows its release in the neutral environment of the small intestine [27]. Acidic poly-(meth)-acrylate copolymers or cellulose and vinyl derivatives are commonly used in phthalates, acetates, trimellitates, succinates, or acetate succinates form. They usually dissolve at pH 5–7 [28] (see Table 2). In the case of ECDDT formation, these polymers are combined with HPMC, with more favorable properties than gelatin for this application, forming the body shell and capsule cap [29].

### 3.2. Gel-Forming Polymers for Enteric Coating

These polymers form a gel layer on the surface of the dosage form when passing through the GIT. The gel layer prevents active pharmaceutical ingredient (API) release from the dosage form in the proximal parts of the GIT (especially in the stomach area). The hydrogel layer is subsequently degraded by two mechanisms, either by spontaneous erosion or by the action of enzymes. Natural substances as polysaccharides (agar, alginates, amylopectin, amylose, arabic gum, arabinogalactan, carrageenan, curdlan, cyclodextrin, dextran esters, dextran’s, gellan gum, glucuronate, guar gum, chitosan, inulin, modified starches, pectin, pullulan, xanthan, xylan, respectively); proteins (collagen, hyaluronic acid); proteoglycans (chondroitin sulfate) [11,28,32,33,34], and others, could be incorporated into the capsule shell in ECDDT technology. Another possibility is to use HPMC [35] or its mixture with natural substances in the ECDDT process [29].

## 4. Related Excipients Used in the Formulation of the Capsule Shell

Film-forming substances from several cellulose derivatives, (meth)-acrylate copolymers, and gelatin create fragile films, therefore, contain plasticizers, most often from several alcoholic sugars, polyols, and their esters or selected polyethers or acylglycerols [36]. In addition, acidic enteric polymers need alkaline substances to dissolve in water, such as ammonia or hydroxides, carbonates, and phosphates of some alkaline metals and earth, which are added to dispersions [37,38]. Gelling substances, like polysaccharides, can be gelled with monovalent or divalent cations of selected alkaline metals and earth [39]. For the easier dispersion of the individual components and the wettability of the finished capsules, the dispersions can also contain several water-soluble and insoluble surfactants [40]. All these components together form homogeneous liquid dispersions, and after drying, solid dispersions in the form of a capsule shell are obtained. The detailed mechanism of action of these substances at the molecular level is described in the literature [7]. The substance groups mentioned above are qualitatively presented in concrete solutions in the overview of inventions in the summary of inventions (see Table A1).

## 5. Brief Description of the Production and Filling of Hard Capsules

In 1846, pharmacist J. C. Lehuby patented two-piece empty hard gelatin capsules, made by dipping metal pins into glycerogelatin. In 1877, another pharmacist, F. A. Hubel, standardized the pins and produced two-part capsules consisting of a body and cap with a size range labeled as 00, 0, 1, 2, 3, 4, and 5 (from the largest size 00 to the smallest size 5) [1]. This labeling of capsules is still used today (see Table 3). In 1931, the company Parke, Davis & Co. (Detroit, MI, USA; now part of Pfizer Corporation, New York, NY, USA) patented the first hard capsule machine [40]. The principle was to prepare a glycerogelatin, with the possible addition of pigment and preservative, and heat it to the required temperature (usually between 45 and 55 °C) [3,7]. A rotating oval pin is then dipped into the prepared, heated solution to ensure uniformity of the application of glycerogelatin. Subsequently, the glycerogel deposition is dried, pulled off from the pin, and trimmed or eventually branded. The capsule bodies and caps are produced separately [3,41]. Later, polymers such as HPMC (the required temperature for hydrogel is approximately 70 °C), pH-dependent, and gel-forming polymers began to be used in the capsule manufacturing process [42,43]. If it is convenient to form an inner and outer shell by coating, immersion could be repeated in different polymer dispersions (double immersion technology). This procedure is preferably used to prepare the outer enteric coating [17]. Until 1968, the capsules had straight, smooth walls, while later conical shapes and various grooves were designed (e.g., Snap-fit (Lock-caps), Eta-lock, Coni-snap, Coni-snap supro capsules) to facilitate the closure of the body with the cap (see Figure 1) [3,40]. Manual and semi-automatic devices could be used for encapsulation in laboratory conditions. API, or its mixture with a suitable filler, is here filled into the capsule bodies and enclosed with a cap [44]. In the industry, automatic devices are used with two leading technologies for filling capsules. The first is tamping (the capsule filling is pressed into the capsule’s body and closed with the cap) [45]. The second technology is damping or dosator (the capsule filling is pressed into a roller, inserted into the capsule’s body, and enclosed by a cap) [46]. Although the detailed capsule production process is not part of this review article, it is already well-known and described in the available literature [47].

## 6. Patent Research of Empty Enteric Capsules

A patent search was conducted from the Google Patents and EspaceNet patent databases using the keywords “enteric”, “acid resistant”, “delayed release”, “hollow”, “empty”, “shell”, “hard”, and “capsule” in various combinations. In addition, the cited patents from each application and related documents offered by the databases were searched. Documents describing the encapsulation of specific drugs by the in situ method were excluded from the final determination. The cited documents do not distinguish between an “application” and a granted “patent”, which allows for maintaining the “priority” necessary to track the development of knowledge and preserve the intent of the review article. The technology description indicates whether it is a single-shell capsule (indicated as Capsule in the table), a double-shell capsule (indicated as Inner and Outer shell), or an Outer shell only. The individual excipients are taken from the patent ‘claims’ in the order listed. Functional groups of excipients, such as film-forming agents, pH-dependent polymers, and gelling agents, are separated by the conjunction “and”. Commas separate individual groups. These functional groups are not directly mentioned in this research but are implied by the preceding text.

It should be noted that the final composition of the capsules does not necessarily contain all the declared substances, but their use and their combination protect the stated inventiveness. However, these are usually inventions companies own for mass production; not all are commercially available. Instead, they preserve compositions and technologies from the use by other manufacturers, which is currently the primary role of patent protection [51]. Fifty-one relevant patents from 1942 to 2022 were selected and sorted diachronically, in Appendix A. Although these technologies have been around since the 1940s, the funnel graph below shows the increase in patent applications in the last decade. The growing demand for encapsulation of acid-labile and biological materials is evident (see Figure 2). The funnel plot also correlates with the incidence of clinical trials, especially in the last 10 years (see Section 8, clinical trials performed with enteric capsules).

## 7. Commercially Hard Capsules Prepared by ECDDT

The capsule shell contains the pH-dependent or gel-forming polymers mentioned above. If the capsules are administered in the FMT process, patients could be premedicated with a proton pump inhibitor, changing the pH in the stomach to 5. However, it could lead to the dissolution of pH-dependent enteric polymers already in the proximal parts of the GIT. Therefore, using capsules made of gel-forming polymers may be preferable here. On the other hand, using pH-dependent polymers avoids premedication [52].

### 7.1. ECDDT Prepared from pH-Dependent Polymers

The first possibility of ECDDT is to manufacture capsule shells from pH-dependent polymers based on acidic cellulose derivatives, for now in phthalates, and acetate succinates forms or from their combinations with HPMC. Based on this principle, the pharmaceutical company CapsCanada (Windsor, ON, Canada) introduced AR Caps^®^ and Capsugel^®^ (Lonza Company, Greenwood, SC, USA) and launched two capsules under the trade name EnTRinsic^TM^ and Vcaps^®^ Enteric.

#### 7.1.1. AR Caps^®^ Capsules

The capsules were introduced in 2013 and have been on the market since 2015 [29]. They are made from HPMC and HPMCP in a ratio of 4:6 [53]. Due to a low moisture content of 4–10% in the capsule shell, the manufacturer recommends them for hygroscopic and hydrolyzing drugs. In addition, capsules are stable over a wide range of air humidity and are not subject to cross-linking [54]. The product information claims that AR Caps^®^ capsules are stable for 60 min in the SGF. Their disintegration begins in the SIF, and capsules are completely released in the duodenum within 60 min [54,55]. Nevertheless, a related study by Al-Tabakha M. et al. shows that AR Caps^®^ capsules filled with paracetamol do not meet the USP requirements for disintegration and dissolution for the delayed-release dosage form. Capsules disintegrate earlier than 60 min in SGF and release more than 15.7% of their content during the 120 min in SGF, and only 46.8% in 45 min in the continuous dissolution test. The study also finds no significant difference in the protection of hygroscopic material (concretely polyvinylpyrrolidone) compared to conventional HPMC capsules [29]. However, AR Caps^®^ capsules were used in a study by Marcial G.E et al. to encapsulate lyophilized bacteria *Lactobacillus johnsonii N6.2* investigated as a potential substance for preventing the development of diabetes, where they provide suitable results for administering biological material to the GIT [56]. Capsules are available in sizes ranging from 000 to 5 [55].

#### 7.1.2. EnTRinsic^TM^ Capsules

The company Capsugel^®^ launched the product in 2015, claiming it was the first capsules prepared by ECDDT with complete enteric protection [57]. The capsule shell is made of pure CAP with a pin dipping process using commercially available machines without any other additive, with 2–7% water content. The thickness of the capsule shell achieves acid resistance. The capsules should dissolve at a pH above 5.5 due to the polymer used for their production [58,59]. According to the manufacturer, capsules fully comply with Ph. Eur., USP, and JP requirements [60]. In a related study, Benameur H. et al. concluded that EnTRinsic^TM^ capsules with esomeprazole meet USP disintegration and dissolution specifications for delayed-release drug form and are bioequivalent to gastro-resistant Nexium^®^ capsules produced by AstraZeneca Plc (Cambridge, UK) [18,61]. In addition, a clinical study by Sager M. et al. investigated the amount of caffeine released from EnTRinsic^TM^ capsules using saliva detection and magnetic resonance imaging (MRI) in vivo after a meal. Results from 14 out of 16 subjects demonstrated the ability to protect the capsule content in the gastric environment because of their disintegration only in the small intestine [62]. Furthermore, capsules were successfully used in the clinical evaluation of lyophilizate-containing bacteria of the human microbiome [63]. Sizes 0 and 3 are commercially available, other sizes could be supplied to order [60].

#### 7.1.3. Vcaps^®^ Enteric Capsules

The capsules were launched in 2015 [64]. They consist of HPMC and HPMCAS, contain less than 6.0% water, and are not subject to cross-linking [65]. The manufacturer, Capsugel^®^, declares that the capsules have been evaluated in vitro and in vivo with various substances, such as paracetamol, dimethyl fumarate, budesonide, or bisacodyl compliantly with USP, Ph. Eur., and JP requirements for delayed release [66]. In addition, Varga A. et al. demonstrated that capsules could be used to encapsulate frozen or lyophilized FMT with results comparable to administration by nasogastric or nasojejunal tubes [52]. Monschke M. et al. also used Vcaps^®^ Enteric capsules to encapsulate amorphous drug dispersions to prevent their disruption in the acidic stomach area, resulting in the delayed release of dosage form during passage in the small intestine [67]. Vcaps^®^ Enteric capsules were also applied for encapsulation of sprayed-dried pectin microparticles with peptides (Lanreotide acetate; Octreotide acetate) with satisfactory results in dissolution testing (total dissolution time of peptides from the capsule after 120 min in the intestinal area) [68]. The manufacturer usually supplies size 0 as standard, with other sizes available to order [69].

### 7.2. ECDDT Prepared from Gel-Forming Polymers

The second principle of ECDDT is producing a capsule shell from a mixture of gel-forming polymers such as pectin or gellan gum, often in combination with HPMC or another formative agent. Various companies worldwide are using this technology to produce their enteric capsule products. For example, company Capsugel^®^ (Lonza Company, Colmar, France) introduced capsules DRcaps^TM^; Suhueng Capsule (Seoul, South Korea) produced EMBO CAPS^®^ AP; BioCaps^®^ (El Monte, CA, USA) launched BioVXR; and the company ACG (Mumbai, India) manufactured capsules under the trade name ACGcaps™ HD.

#### 7.2.1. DRcaps^TM^ Capsules

In 2011, Capsugel^®^ launched DRcaps^TM^ capsules consisting of HPMC and gellan gum (a heteropolysaccharide of glucose, rhamnose, and glucuronic acid containing acidic functional groups, allowing pH-dependent dissolution [70]) in the ratio of 95:5 [71]. The water content in their structure is 4–7% allows, according to the manufacturer’s encapsulation of hygroscopic drugs [72]. While conventional hard gelatin capsules disintegrate approximately 5 min after administration in SGF [73], DRcaps^TM^ starts to disintegrate 45 min later in the dissolution test. Besides, according to an in vivo scintigraphy test made by the producer, complete disintegration occurs in the small intestine after 20 min [74]. According to Al-Tabakha M. et al., DRcaps^TM^ capsules with paracetamol do not meet the USP requirements for disintegration and dissolution tests because they disintegrated in SGF within 60 min. In the dissolution test, capsules released 19.6% of API in SGF after 120 min, and subsequently, in SIF, only 34.2% within 45 min [29]. A similar result was reached by Grimm M. et al., with even a third less capsule content released in SIF [75]. DRcaps^TM^ capsules were also used in a study by Youngster I. et al. as a prospective dosage form in FMT. DRcaps^TM^ capsules were filled with fecal material with trypan blue as a dye and deep frozen. The capsules began to release dye after 115 min at pH ≥ 3 [76]. A longer time was achieved by inserting capsule size 0 into capsule size 00 (so-called Cap-in-Cap system). The total disintegration time in SGF ranged from 148 ± 42 to 168 ± 38 min depending on the capsule’s contents. MRI then confirmed the gastric passage at fasting and API release in the small intestine [75]. In a study by Marzorati M. et al., DRcaps^TM^ capsules containing encapsulated bacterial strain *Bifidobacterium longum* or acid-labile enzymes were used. During an in vitro dissolution test, the capsules showed satisfactory results in capsule content protection when they stayed intact in the stomach area and disintegrated entirely at the beginning of the simulated small intestine. Additionally, DRcaps^TM^ capsules preserved bacterial viability in the capsule even after long-term storage [77]. Classic HPMC capsules and DRcaps^TM^ were compared by the Cap-in-Cap method in different combinations. Only doubled DRcaps^TM^ capsules showed pharmacopeial acid resistance [78]. However, significant protection of the hygroscopic drug was not demonstrated after comparison with conventional HPMC capsules [29]. The commercially available sizes of DRcaps^TM^ capsules are from 000 to 4 [79].

#### 7.2.2. EMBO CAPS^®^ AP Capsules

The capsules appeared on the market in 2018. They consist of HPMC and pectin, with 5–7% water content [80]. According to the manufacturer, they release 12.0% after 120 min in SGF and more than 90.0% of the unspecified API after 30 min in SIF [81]. However, Grimm M. et al. discovered that capsules in the dissolution test released over 20.0% of paracetamol as their content during 120 min in SGF and subsequently more than 90.0% in 45 min in SIF [75]. These results do not comply with the limit requirements of the USP, so they cannot be considered enteric. The producer declares their use for encapsulating probiotics and hygroscopic materials [56] and it is already used for encapsulating bacterial lyophilizates and plant extracts in market products [82]. They are available in the size range of 00EL–4 [83].

#### 7.2.3. BioVXR^®^ Capsules

BioVXR capsules were patented in 2016. A producer, the company BioCaps^®^, does not fully disclose the composition of its product, except for HPMC [75]. However, the associated patent mentions using polymers such as pectin, propylene glycol, alginate, or xanthan gum and gelling agents such as gellan gum or carrageenan [84]. According to the producer, BioVXR^®^ capsules are fully acid-resistant, with no more than 6.95% of the capsule content leaked in 120 min in SGF during the dissolution test, then in SIF within 45 min 89.1% of unspecified API [85]. A mixture of HPMC, gellan gum, and pectin roughly corresponds to this dissolution profile according to the patent composition [84]. Yet, results from the dissolution test provided by Grimm M. et al. showed a drug release of 48.5% from BioVXR^®^ capsules at 120 min in SGF, and subsequently in SIF less than 80.0% after 45 min, which does not meet the requirements of the delayed-release drug form [75]. Capsules are available in sizes 00, 0, and 1 [86].

#### 7.2.4. ACGcaps^TM^ HD Capsules

The capsules, consisting of HPMC, with other components not specified, were introduced in 2019 as a part of the company’s HPMC capsule line [87,88]. According to the manufacturer, the capsules disintegrate by a dual mechanism, i.e., also depending on the transit time through the GIT [89], and could be used for encapsulation hygroscopic drug due to their low moisture of capsule shell and resistance to cross-linking [90]. In addition, the product information states solubility at pH > 5.5 [91]. A study by Ashish V. et al. examined ACGcaps^TM^ HD capsules with lyophilized or oil-based content of *Lactobacillus casei* for 120 min in SGF, and then for 360 min in SIF at an electromagnetic stirrer. Viability detected by in vitro cultivation of the collected samples after tests showed no significant release of capsule content in SGF, while 70.0–90.0% was released in SIF [87]. However, this is not a pharmacopoeial method for examining delayed-release drug forms. Nevertheless, these capsules are more suitable for lyophilized bacteria *Lactobacillus fermentum* compared to conventional gelatin capsules [35]. The ACG company offers capsules in sizes from 000 to 5 [89,90].

The individual commercially manufactured enteric capsules, their manufacturer, quality composition and the pharmacopoeial assumptions mentioned in the text above are summarized in Table 4.

## 8. Clinical Studies Performed with Enteric Capsules

A search for clinical studies was conducted from Google Scholar, ScienceDirect, and Scopus using the keywords “enteric”, “acid resistant”, “delayed release”, “capsule product name”, “capsule”, “clinical”, “individual treatment”, “trial”, “study”, and in various combinations. A list of clinical studies is presented in correlation with a funnel plot to show the success of introducing this enteric dosage form in recent years (Table 5; Figure 2). According to a recent clinical systematic reviews and meta-analyses, a series of studies with enteric capsules have been conducted [13,92]; however, it is not always possible to identify the specific enteric capsule type and its manufacturer. Table 5 summarizes those clinical trials for which this information (capsule type, manufacturer) has been published.

In clinical or preclinical trials, for documentation and a better understanding of the capsule behavior after its administration to the GIT, non-invasive imaging techniques are used. Techniques such as MRI [75], single-photon emission computed tomography (SPECT) [106] and X-ray imaging could track movement through the tract [107]. Of additional imaging techniques, computed tomography (CT) allows 3D imaging of soft tissue structures. In combination with positron emission tomography (PET), CT could investigate the in vivo behavior of capsules in the GIT [108] (see Figure 3).

## 9. Conclusions

Over the last decade, capsules prepared by ECDDT have started to appear on the worldwide market. Their shells are formed by synthetic and semi-synthetic pH-dependent polymers, which prevent dissolution in the acidic stomach environment or swelling natural or semi-synthetic polymers, which gel-forming properties extend the disintegration of drug form into the small intestine area. Although their commercial availability allows industrial manufacturing and preparation of drugs under laboratory conditions in pharmacies, pharmacopoeial limits for acid resistance may only sometimes be achieved with their use. Nevertheless, all capsules from the manufacturers mentioned above could help protect the encapsulated material from the acidic stomach environment. In addition, preventing release in the gastric environment can be enhanced by repeatedly encapsulating the drug capsules in a larger capsule (Cap-in-Cap). At the same time, the limits of pharmacopeia are not always guaranteed in pH 6.8 conditions. Another complication is the encapsulation of frozen FMT, which thaws during GIT passage. Due to the neutral environment, the capsule shell melts in the stomach, causing damage to the FMT.

Furthermore, specifically, when preparing a filling made of FMT, ethical problems arise on the part of patients. Currently, there are several trends to overcome the previous complications. One possibility is the application of an internal insoluble layer, which would prevent the capsule from dissolving from the inside, but which would disintegrate due to the impact of intestinal motility. Another developing option is manufacturing capsules directly in hospital pharmacies using 3D printing, allowing, for instance, a more precise adjustment of the release according to the thickness of the capsule wall. Finally, the cultivation of bacterial repopulation consortia is beginning to be considered in practice. It would solve the ethical aspect of FMT encapsulation by replacing it with selective cultures grown ex vivo.

## Figures and Tables

**Figure 1 pharmaceuticals-15-01398-f001:**
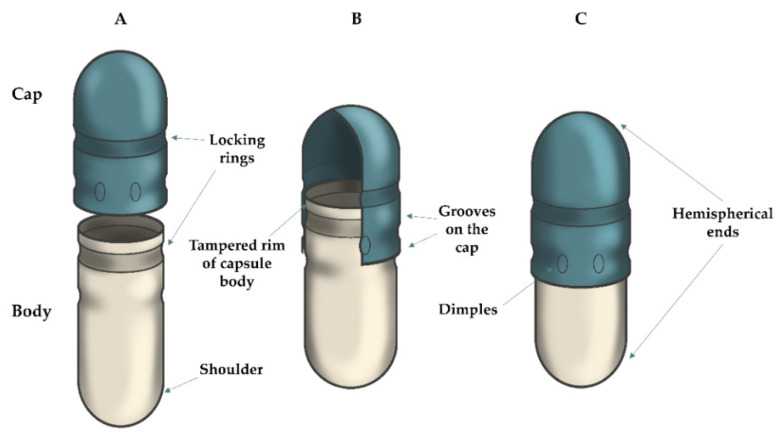
Structure of the common type of enteric hard capsule prepared with ECDDT (Snap-fit type): (**A**) open position with two capsule parts (cap, body); (**B**) pre-closed position; (**C**) closed position of hard capsule.

**Figure 2 pharmaceuticals-15-01398-f002:**
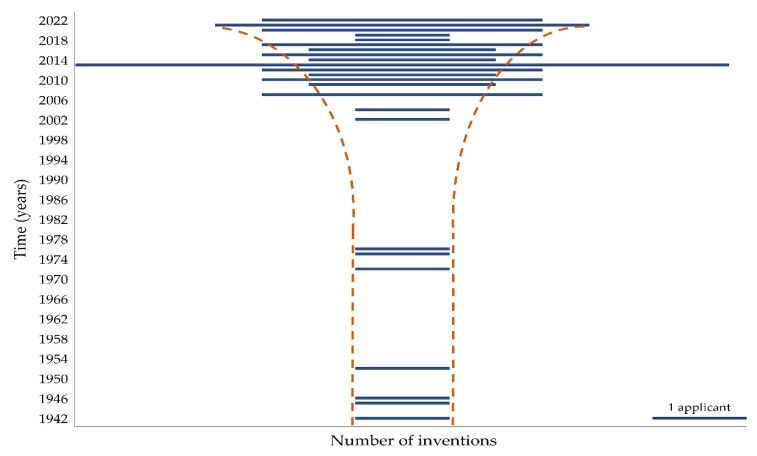
The funnel plot. Approximate growth of patent applications in time.

**Figure 3 pharmaceuticals-15-01398-f003:**
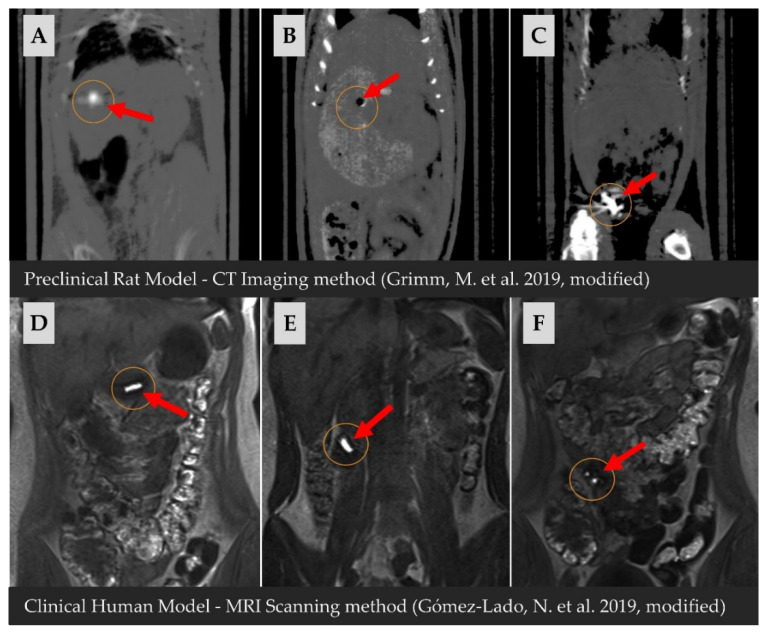
Documentation of the behavior of the enteric capsule after administration into the organism with two methods (MRI; CT) [75,108]; preclinical study performed on multiple rat models (differences in transit time could occur): (**A**)—capsule in the stomach after administration, (**B**)—disintegrated capsule in the stomach after 540 min, (**C**)—disintegrated capsule in caecum after 180 min; clinical study performed on one human model: (**D**)—capsule in the stomach after 5 min, (**E**)—capsule in ileum after 150 min, (**F**)—disintegrated capsule in ileum after 210 min. Reprinted from Eur. J. Pharm. Sci., 129. Grimm, M.; Ball, K.; Scholz, E.; Schneider, F.; Sivert, A.; Benameur, H.; Kromrey, M.L.; Kühn, J.P.; Weitschies, W. Characterization of the Gastrointestinal Transit and Disintegration Behavior of Floating and Sinking Acid-Resistant Capsules Using a Novel MRI Labeling Technique, 163–172, Copyright (2019), with permission from Elsevier.

**Table 1 pharmaceuticals-15-01398-t001:** Conditions and limits in three different pharmacopeias (European, Unites States, Japanese) [19,22,23,24,25,26].

Pharmacopoeia/Abbreviation	European Pharmacopoeia/Ph. Eur.		United States Pharmacopoeia/USP		Japanese Pharmacopoeia/JP	
**A. Disintegration test**						
**Conditions ^1^**	1.	2.	1.	2.	1.	2.
Time (min)	120	60	60	MON	120	60
Disks	No	Yes	MON	MON	No	Yes
Limits (units)	0/6	6/6	0/6	MON	0/6	6/6
**B. Dissolution test**						
**Conditions ^1^**	1.	2.	1.	2.	1.	2.
Time (min)	120	45/MON	120	45/MON	120	MON
Limits (%)	≤10	Q + 5	≤10	Q + 5	≤10	Q_MON_ + 5
Pharmacopoeia procedure ^2^	Method A Continual two-step dissolution		Method A Continual two-step dissolution		If not specified, proceed with both stages of the test separately	
	Method B Separated two-step dissolution		Method B Separated two-step dissolution			

^1^ Experimental conditions in disintegration and dissolution tests (first (1.) in pH 1.2; second (2.) in pH 6.8); ^2^ Two methods for proceeding with the dissolution test (A—Placed first in acid stage, then add the required amount of buffer to an acid solution; B—First placed in acid stage, then replaced to buffer stage); MON—Specified in individual pharmacopeial monographs; Q—the dissolution end time (total dissolution time) given by the general lower limit (amount of released API 75.0% if not specified in the individual pharmacopeial monograph); Q_MON_—Amount of released API specified in the monograph.

**Table 2 pharmaceuticals-15-01398-t002:** Overview of the most common pH-dependent enteric polymers [28,30,31,32].

Commercial Name/Abbreviation ^1^	Chemical Name	Solubility at pH Values ^2^
Eudragit^®^ FS	A copolymer of methyl acrylate, methyl methacrylate, and methacrylic acid in a ratio of 7:3:1	7.0
Eudragit^®^ S	A copolymer of methacrylic acid and methyl methacrylate in a ratio of 1:2	7.0
Eudragit^®^ L	A copolymer of methacrylic acid and methyl methacrylate in a ratio of 1:1	6.0
Eudragit^®^ FL	A mixture of a copolymer of ethyl methacrylate and methyl methacrylate with a copolymer of methacrylic acid and ethyl methacrylate in a ratio of 1:1	5.5
CAS	Cellulose acetate succinate	5.8–6.2
CAP	Cellulose acetate phthalate	5.5–6.2
CAT	Cellulose trimellitate	5.2–5.5
HPMCAS	Hydroxypropyl methylcellulose acetate succinate	5.5–6.8
HPMCP	Hydroxypropyl methylcellulose phthalate	5.0–5.5
PVAP	Polyvinyl acetate phthalate	5.0–5.5

^1^ Abbreviation of enteric polymers are used in a whole article according to this table; ^2^ This is the value at which the polymer dissolves; this could vary for some substances depending on the ratio of substituents, with literature values often differing slightly.

**Table 3 pharmaceuticals-15-01398-t003:** Sizes, volumes, weights, dimensions of capsule parts (body, cap), and the whole capsule [48,49,50].

Size	000	00EL	00	0EL	0	1EL	1	2EL	2	3	4	5
Body length (mm)	22.2	22.2	20.2	20.2	18.4	17.7	16.6	16.7	15.3	13.6	12.2	9.3
Cap length (mm)	12.9	12.9	11.7	11.7	10.7	10.5	9.8	9.7	8.9	8.1	7.2	6.2
Capsule length (mm)	26.1	25.3	23.3	23.1	21.7	20.4	19.4	19.3	18.0	15.9	14.3	11.1
Weight (mg) ^1^	163	130	118	107	96	81	76	66	61	48	38	28
Volume (mL)	1.37	1.02	0.91	0.78	0.68	0.54	0.50	0.41	0.37	0.30	0.25	0.13

^1^ Some manufacturers list the same weight of gelatin and HPMC capsules.

**Table 4 pharmaceuticals-15-01398-t004:** Overview of commercially produced capsules using ECDDT.

Product Name	Manufacturer/Country of Origin	Composition of Capsule	Pharmacopoeial Acid Resistance
A. Capsules prepared from pH-dependent polymers			
AR Caps^®^	CapsCanada/Canada	HPMC, HMPCP	No [29]
EnTRinsic^TM^	Capsugel^®^/USA	CAP	Ph. Eur., USP, JP [60]
VCaps^®^ Enteric	Capsugel^®^/USA	HPMC, HPMCAS	Ph. Eur., USP, JP [66]
B. Capsules prepared from gel-forming polymers			
DRcaps^TM^	Capsugel^®^/France	HPMC, gellan gum	No [29]
EMBO CAPS^®^ AP	Suhueng Capsule/ Republic of Korea	HPMC, pectin	No [75]
BioVXR^®^	BioCaps^®^/USA	HPMC, gellan gum/pectin ^1^	No [75]
AGCcaps^TM^ HD	ACG/India	HPMC ^2^	Not specified

^1^ Derived composition based on an associated patent [84]; ^2^ other components are not listed.

**Table 5 pharmaceuticals-15-01398-t005:** Overview of clinical studies with enteric capsules arranged in chronological order.

Clinical Study	Type of Capsule	Specification of Dosage Form	Single Dose/A Total Dose of Capsules	Capsule Content	Clinical Indication	Patient Population	Clinical Success Rate
Walker, E.G., et al., 2022 [93]	DRcaps^TM^	-	2/2	Bitter extract of *Humulus lupulus*	Regulating Energy Intake (EI)	19	EI ↓ 17.54%
Reid I. et al., 2022 [94]	Vcaps^®^ Enteric	One capsule with 20 mg of API	1; 2; 3/NA	Zoledronic acid (ZA); ZA as microparticles	Safety in bone resorption	5	Positive after one week
Zain N. et al., 2022 [95]	DRcaps^TM^	-	5/5	Lyophilized bacteria in FMT	rCDI ^1^	7	86%
Varga A. et al., 2021 [52]	Vcaps^®^ Enteric	-	5–11/NA	Various parts of fecal suspension	rCDI	28	82.14%
Reigadas E. et al., 2020 [96]	DRcaps^TM^	Cap-in-Cap system	4–5/NA	Lyophilized bacteria in FMT	rCDI	32	87.5%
Leong K. S. W. et al., 2019 [97]	DRcaps^TM^	Cap-in-Cap system ^2^	14/28	Frozen bacteria in FMT	Obesity/insulin resistance in adolescent	80	NA
Jiang Z., et al., 2018 [98]	AR CAPS^®^	-	NA/27	Lyophilized bacteria in FMT	rCDI	31	84%
Chehri M. et al., 2018 [99]	DRcaps^TM^	Cap-in-Cap system	25 (daily)/75	Frozen bacteria in FMT	rCDI	9	88.9%
Halkjær S. et al., 2018 [100]	DRcaps^TM^		25 (daily)/ 12 days	Frozen fecal in FMT	Irritable Bowel Syndrome (IBS)	52	36.4
Staley C. et al., 2017 [101]	DRcaps^TM^	Cap-in-Cap system	2–4/NA	Lyophilized bacteria in FMT	rCDI	49	88%
Khana S. et al., 2016 [102]	DRcaps^TM^	-	15/30 1–12/NA	Various numbers of bacteria	rCDI	15 15	80% 86.7%
Hirsch B. et al., 2015 [103]	DRcaps^TM^	Cap-in-Cap system + gelatin capsule ^3^	6–22/NA	Frozen bacteria in FMT	rCDI	19	89%
Youngster I. et al., 2014 [76]	DRcaps^TM^	Cap-in-Cap system	NA/30	Frozen bacteria in FMT	rCDI	20	90%
Jones M. et al., 2012 [104]	DRcaps^TM^	-	2 (daily)/126	Lyophilized bacteria *Lactobacillus reuteri NCIMB 30242*	Hypercholesterolemia	131	LDL-C ^4^ ↓ 11.64%
Jones M. et al., 2012 [105]	DRcaps^TM^	-	2 (daily)/126	Lyophilized bacteria *Lactobacillus reuteri*	Clinical safety of *Lactobacillus reuteri*	131	Positive; Not quantified

^1^ System of two capsules encapsulated into each other; ^2^ recurrent clostridium difficile infections; ^3^ system of two capsules encapsulated into each other + gelatin capsule as a third cover layer; ^4^ low-density lipoprotein-cholesterol. NA—not applicable.

## Data Availability

Not applicable.

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
