# Peer review of "Commercially Available Enteric Empty Hard Capsules, Production Technology and Application"

_pharmaceuticals, 2022, doi:10.3390/ph15111398_

Round 1

Reviewer 1 Report

The present manuscript addresses the commercially available enteric empty hard capsules suitable for individualized treatment. This manuscript provides critical contributions to oral delivery. However, this manuscript should be deemed acceptable for publication after the major revision in accordance with the suggestions listed below.  

1.     Authors have ignored large portions of the recent literature and other important work on pharmacokinetic studies. The authors should be more precise here.

2.     In Title “individualized treatment” claimed.  Authors failed to discuss Enteric capsules for individualized disease treatment, Crohn's disease, or colorectal cancer.

3.     Section 2. Enterosolvency of solid dosage forms: Provide one figure on oral enteric dosage forms that resist the acidic environment in the stomach and then dissolve in an environment of the small intestine

4.     Include a list of patents on enteric empty hard capsules

5.     Most of the statements are not cited with relevant references. The statements need to validate by references.

6.     Future prospective should be discussed in the conclusion section. 

Author Response

Dear reviewer, thank you for your time. We have tried to reflect your comments to the best of our ability. Since your comments correspond with those of the other reviewer, for your better orientation, we are enclosing a shared document.

Reviewer 2 Report

I reviewed the review manuscript entitled: “Commercially available enteric empty hard capsules suitable for individualized treatment”. I found the topic extremely interesting. The quality of the literature review is very good. 

Here are some comments and curiosities I have about the manuscript.

If it is possible in this journal, I would put a summary table of abbreviations only at the beginning of the review before the introduction section. It will make it easier to find the meaning of the abbreviations and to make the document easier to read. 

ABSTRACT

Line 11: I am not sure if "an elegant method" is an adequate way to describe it. It sounds strange to me. 

Line 19: I would change “the article” for “this literature review”.

Line 20: I would remove the last sentence “A similar review has not yet been published”. 

In the keywords I would change “commercial” for “commercial capsules”

INTRODUCTION

Line 26: Only 18% are capsules? A reference should be added. 

ENTEROSOLVENCY OF SOLID DOSAGE FORMS

Line 64: in reference to the small intestine, is a distinction made between the upper and lower small intestine or is the entire small intestine as a unit?

Line 66: the pH value of 5 to 6 for the stomach, is this variability in the values genetic or is it due to people with previous pathologies?

Line 83-84: I don't know the "disks" method, could you explain it briefly?

ENTERIC POLYMERS

3.1. Section: If possible, I would give practical examples of some of the common treatments currently used with pH-dependent enteric polymers.

3.2. Section: If possible, I would give practical examples of some of the common treatments currently used with gel-forming polymers for enteric coating.

COMMERCIALLY HARD CAPSULES PREPARED BY ECDDT

Line 163: Capsugel® is an ECDDTT made form gel-forming polymers, and you discuss it in the next section, so I would remove it from here. 

Line 201: are these capsules currently being used (without being clinical trials) for pathologies of the human microbiome?

Line 208-209: as in the previous section, it would be very interesting to know if it possible if they are currently being used in this type of treatment without being clinical trials. 

Line 280: in vivo should be in cursive. 

CONCLUSIONS

Conclusions should remark something about the future challenges or the need of more research in this field. Maybe that more research or in vivo trials are needed to corroborate the efficacy of different ECDDT capsules with different fillers, as there are notable differences between encapsulating chemical agents and freeze-dried microorganisms. And there are also significant differences between the values of disintegration and dissolution of the capsules provided by manufacturers and the results obtained from independent researches, as you cited in the manuscript. 

Author Response

(The authors gave the same response as above.)

Round 2

Reviewer 1 Report

The authors have revised the manuscript as per the suggestions. I recommend accepting the manuscript as it is.